# Synthesis under Normal Conditions and Morphology and Composition of AlF_3_ Nanowires

**DOI:** 10.3390/nano13172413

**Published:** 2023-08-25

**Authors:** Albert Dautov, Kotstantin Kotlyar, Denis Butusov, Ivan Novikov, Aliya Khafizova, Artur Karimov

**Affiliations:** 1Faculty of Electronics, St. Petersburg State Electrotechnical University “LETI”, 197376 Saint Petersburg, Russia; ianovikov@stud.etu.ru (I.N.); amkhafizova@stud.etu.ru (A.K.); 2Faculty of Physics, St. Petersburg State University, Universitetskaya Embankment 13B, 199034 Saint Petersburg, Russia; konstantin21kt@gmail.com; 3Department of Physics, Alferov University, Khlopina 8/3, 194021 Saint Petersburg, Russia; 4Institute for Analytical Instrumentation RAS, Rizhsky 26, 190103 Saint Petersburg, Russia; 5Computer-Aided Design Department, St. Petersburg Electrotechnical University “LETI”, 5 Professora Popova St., 197022 Saint Petersburg, Russia; dnbutusov@etu.ru (D.B.); aikarimov@etu.ru (A.K.); 6Youth Research Institute, St. Petersburg Electrotechnical University “LETI”, 5 Professora Popova St., 197022 Saint Petersburg, Russia

**Keywords:** AlF_3_ nanowires, hydrofluoric acid, metal fluorides

## Abstract

AlF_3_ has interesting electrophysical properties, due to which the material is promising for applications in supercapacitors, UV coatings with low refractive index, excimer laser mirrors, and photolithography. The formation of AlF_3_-based nano- and micro-wires can bring new functionalities to AlF_3_ material. AlF_3_ nanowires are used, for example, in functionally modified microprobes for a scanning probe microscope. In this work, we investigate the AlF_3_ samples obtained by the reaction of initial aluminum with an aqueous hydrofluoric acid solution of different concentrations. The peculiarity of our work is that the presented method for the synthesis of AlF_3_ and one-dimensional structures based on AlF_3_ is simple to perform and does not require any additional precursors or costs related to the additional source materials. All the samples were obtained under normal conditions. The morphology of the nanowire samples is studied using scanning electron microscopy. We performed an intermediate atomic force microscope analysis of dissolved Al samples to analyze the reactions occurring on the metal surface. The surface of the obtained samples was analyzed using a scanning electron microscope. During the analysis, it was found that under the given conditions, whiskers were synthesized. The scale of one-dimensional structures varies depending on the given parameters in the system. Quantitative energy-dispersive x-ray spectroscopy spectra are obtained and analyzed with respect to the feedstock and each other.

## 1. Introduction

Metal fluorides are promising for catalysis applications [1,2,3,4,5], and are often used in C-H and C-F bond activation reactions or hydroarylation reactions, as ionic conductors [6] and luminescent materials [7,8,9,10], applying sol-gel technology with trifluoroacetic acid as a source of fluorine. In addition, ref. [6] reports the atomic layer deposition of LiAlF_4_ film, which has high conductivity, homogeneity, chemical stability, and electrochemical inertness. AlF_3_-based glasses demonstrate enhanced chemical and thermal stability and excellent moisture resistance, and are successfully used in infrared lasers [11].

One of the most important representatives of binary compounds, MeF3, is aluminum fluoride (AlF_3_), which is an inorganic crystalline substance under normal conditions. Due to its electro-physical properties, it finds applications as a catalyst in organic synthesis, as an additive for electrolytes for lowering melting points, as flux for oxide separation and removal [12], and as an element of multilayer optical structures [13]. It also exhibits excellent properties as a material for the fabrication of far-infrared mirrors [14], in which this trifluoride is used as a protective layer for Al mirrors. In multilayer optical structures, AlF_3_ is a component of the AlF_3_/ThF_4_ heterostructures, which are superior to the LiF/ThF4 optical structures in terms of their optical parameters. The authors of the work [11] provide a comparison of the two structures in terms of such parameters as solar light transmission coefficient, solar light reflection coefficient, etc., and show that it outperforms LiF/ThF4 structure. It is also a promising candidate for fiber lasers in the mid-infrared region [13]. It is known that the use of aluminum fluoride in the AlF_3_/SiO_2_/Si system allows one to localize a high density of fixed negative charges in fluorine vacancies, serving as a source of a strong drift field [15]. AlF_3_/polyimide films increase dielectric permeability as well as reduce leakage currents compared to conventional polyimide films [16]. Interestingly, AlF_3_ can also be used to prevent tooth discoloration [17]. AlF_3_ can be used as a film to cover the electrode surfaces of lithium-sulfur batteries to prevent the “shuttle effect”, which results in the transfer of polysulfide and the formation of dendrites. This leads to a sharp decrease in productivity and low kiln efficiency [18]. The use of AlF_3_ as a surface coating improves the electrochemical characteristics of K_1.39_Mn_3_O_6_ microspheres, which, in turn, has a beneficial effect on the cyclic stability, reversible capacity, and high-speed capacity of batteries. The stabilized structure of AlF_3_ can play the role of a physical isolator between the active substances and electrolytes to prevent the occurrence of adverse reactions and inhibition of transition metal ions in batteries [18]. In most of these applications, an increased surface-to-volume ratio is expected to increase the performance of AlF_3_ material, which is why a cost-effective synthesis technology of one-dimensional AlF_3_ structures is highly desirable.

Synthesis of AlF_3_ in the form of one-dimensional (1D) structures, such as nanorods, nanowires (NWs), and nanotubes, enables a high surface-to-volume ratio. This makes 1D AlF_3_-based structures promising for use in supercapacitors and nano-probes of different types [19,20,21,22,23]. There are several methods for producing 1D structures, including AlF_3_-based NWs [24,25,26,27,28,29,30], each characterized by a complicated technological process. Particularly, it is difficult to obtain AlF_3_ without the use of aggressive hydrofluoric acid HF. One of the most popular methods for synthesizing AlF_3_ nanoparticles is the fluorolitic sol-gel synthesis [31], which can be described by the system of equations
M-OR + H-F → M-F + ROH(1)
nM-F → (-N-F-M-F) n/2(2)

In the first reaction, water as a reactant is completely replaced with hydrogen fluoride, which leads to the formation of M-F bonds. The second reaction, if performed correctly, leads to the formation of either metal oxides or nanoscale metal fluorides. Fluoride ions tend to form stable bonds. Consequently, there are not many examples of crystalline fluorides containing fluorine at the ends. One problem with solution-based synthesis methods is that in such reactions we face a system depending on several parameters including time, solution saturation, and amount of solution. In addition, when synthesizing metal fluorides, it should be taken into account that they have a high lattice energy, which makes it difficult to form regular crystal structures [32]. Helical dislocations are a stimulating factor for the growth of one-dimensional anisotropic structures, which provide the synthesis environment with self-preserving helical steps [25]. On top of that, the existing technological methods require relatively high temperatures (often about 450–500 °C), as well as additional raw materials for the intermediate reactions.

As follows from the promising nature of 1D AlF_3_ for use in a large variety of applications, developing a more simple method to synthesize AlF_3_ NWs is desired. The newly proposed production technology should ensure the repeatability of synthesized one-dimensional structures, should not require extreme parameters of the medium, and it is preferred that this method is realized under normal conditions. The current paper presents the synthesis method for AlF_3_ NWs satisfying all these conditions and reports the results of the morphological and compositional characterization of the obtained NWs. These NWs are synthesized in the direct reaction of hydrofluoric acid solution with aluminum under normal conditions. Two cases are considered: a synthesis reaction with food-grade aluminum used for crockery and household appliances, which includes a small proportion of additional components, as well as a reaction with aluminum of a purity of 99.99%. The morphology and composition of the NW samples are studied versus the concentrations of hydrofluoric acid reacting with pure aluminum.

## 2. Design of Experiments

The experiments were carried out as follows. Two types of aluminum alloy were investigated. The first was the food-grade aluminum taken from a 1.7 L pot by KALITVA COOK and served as a raw material for the study (Al_f_). In the analysis of the alloy using EDX method, the following composition was determined: Al = 98.95%, Si = 0.67%, Ca = 0.05%, Fe = 0.21%, Ni = 0.13%, and Zn—0.02%. Pieces of this raw aluminum alloy were cut into 20 × 20 mm samples (plates of 1 mm thickness, and exposed to 70 mL of the hydrofluoric acid (“Sigma Tech”, Vladivostok, Russia) of different concentrations, see Table 1.

The second sample under investigation was the pure aluminum (Al_p_) in the form of pellets, with a nominal pellet mass of m = 2.435 g. This aluminum of grade N4 was produced under technical conditions ETO.021.051 and contains a mass ratio of additives of 0.015% or less. In the second group of experiments, a single pellet was exposed to 40 mL of the hydrofluoric acid of different concentrations, see Table 2. 

The course and rate of the reaction transpired to be dependent on the type of the investigated sample. For the food-grade aluminum alloy, the entire surface of the sample was rapidly covered with a white film at the beginning of the reaction, and then gradually accumulation of hydrogen bubbles became noticeable in the volume. Twenty-five minutes later, the reaction transitioned to the active stage, with fast outgassing. Two hours later, the sample dissolved completely. The solution was left to await precipitation. Then, the reaction was left for twenty-four hours, and the obtained product was treated with 2-methanol propane to prevent contact with the solution residue. In a case of pure aluminum, the reaction was relatively calm with uniform emission of hydrogen. Like in the first experiment, the reaction was left for twenty-four hours after beginning to ensure the metal was fully dissolved and the remains of the hydrofluoric acid were vaporized. As the result of all reactions, we obtained white or transparent crystals with a characteristic odor, resembling table salt crystals in brittleness. We hypothesize that in the case of food-grade aluminum alloy, some additional elements, particularly zinc, may have served as a catalyst to increase the rate of the aluminum dissolution reaction.

The morphology and composition of the original aluminum and synthesized samples were studied using a scanning electron microscope (SEM) Zeiss Supra 25 (Carl Zeiss, Gena, Germany), equipped with an energy dispersive X-ray analyzer (EDX) Ultim Max 100 (Oxford Instruments, Abingdon upon Thames, UK). An intelligent microscope, NT-MDT NTEGRA THERMA (Zelenograd, Russia), was used for atomic force microscope (AFM) analysis.

## 3. Results

In the first experiment, it was assumed that after a reaction between hydrofluoric acid and an aluminum sample, AlF_3_∙H_2_O was obtained, while both reactants and the conditions of the reaction sufficiently differ from those reported in the literature. One of the best-known methods of obtaining aluminum fluoride is neutralizing hydrofluoric acid with aluminum hydroxide 3HF + AL(OH)_3_ = AlF_3_ + 3H_2_O at 90–95 °C applied for one hour. In the course of the reaction, aluminum fluoride crystallizes as AlF_3_∙H_2_O, after which the precipitate is dehydrated at 350 °C [33]. It is possible that the additives in the food-grade aluminum alloy served as a catalyst for the reaction. However, this reasoning was not confirmed by a series of tests with Al_f_ pellets.

According to the AFM results obtained, exposure to HF disturbed the integrity of the original surface, forming a stepped-grid structure, shown in Figure 1b. As can be seen, the flaking of the structure occurred uniformly, regardless of variations in the height of the original relief, as seen in Figure 1a. Based on the assumption that the destruction of the sample material occurred under the catalytic effect of a certain alloy constituent, we can conclude that its greatest concentration was in the places under the “steps” where dissolution was the most intense.

A series of tests with hydrofluoric acid of different concentrations was carried out. As the concentration of hydrofluoric acid in the solution increased, the initial structure of the resulting precipitate changed. During the reaction of sample 6.1 with 8% solution, the precipitate obtained during the reaction copied the shape of the container in which the experiment took place in Figure 2b. The sample has a sandwich structure: a transparent layer in the middle and a white salt-like surface. In the experiment with sample 1.1, the precipitate fell out as flakes. In addition to the precipitate, a thin layer of unreacted Al remained in the container. It is interesting that samples 1.1–1.2 transpired to be unstable and after two weeks acquired a macro-porous structure, as shown in Figure 2a. The chemical composition of the obtained samples was investigated using EDX, while SEM visualization represented the morphology of the output structures. Figure 2 shows a comparative analysis of the spectra of the initial aluminum (Figure 3, “Metal”) and the studied substance, which is the result of the reaction (Figure 3, “Solid”). According to the peaks, we can conclude that the resulting material consists mainly of Al and F. The calculated atomic percentages of different elements are the following: F = 71.17%; Al = 29.53%; Si = 0.57%; Fe = 0.08%; Ni = 0.08%; and Zn = 0.01%.

Figure 4 shows the images obtained by scanning SEM during the analysis of sample 6.1. One can see from Figure 4 that the surface of the sample consists of an ensemble of 1D microwires. The average height of 1D structures is 4 µm, the length of the hexagon side is 0.82 µm, and the surface density is in the order of $10^{4}$ × 1/cm^2^. EDX was also used to determine the microwire composition.

The following analysis revealed that the percentage of Zn in the microwires was higher than in the matrix as a whole. It is interesting that, depending on the concentration of the solution of HF with which the initial material interacted, the size of the wires and their geometry can change. For example, Figure 5 shows an SEM image of sample 5.1, where the NWs are thinner than in sample 6.1. The average height of these NWs is 1 µm, the length of the hexagon side is 0.26 µm, and the surface density is in the order of $3\times 10^{4}$ 1/cm^2^.

The SEM of sample 4.1 is given in Figure 6. The average height of these NWs is 4 µm, the average diameter is around 0.4 µm; and the surface density is in the order of $4\times 10^{4}$ 1/cm^2^.

During the study of the formation of 1D structures, an interesting fact was discovered. It transpired that NWs did not form in the reaction with maximum concentration. Thus, in the synthesis of sample 1.1, no NWs were formed, and the microstructure of the material is a rough surface.

To further clarify and strengthen the results, we performed a series of experiments with pure Al (Al_p_) samples. Table 2 presents the experimental data (in the synthesis of sample 5.2 the mass of the initial aluminum was lower than average, which may have affected the overall dependence of the given ratios). Figure 7 shows the SEM images of the obtained samples. The data of the Al_f_ and Al_p_ reactions appear very similar.

The sample in Figure 7a was obtained during the reaction of Al_p_ with 16.3% solution. The geometry of 1D structures is similar to that shown in Figure 4, and 29.4% solution gave rise to the structures shown in Figure 7b resembling NWs in Figure 5. According to the SEM data, we can also say that the obtained samples exhibit the ability to accumulate charge. We clearly observe a structural transformation of the surface, from a completely smooth surface with the presence of local dislocations to a complete coverage of the surface with NWs.

The difference can be also found in the resulting macrostructure. In the 16.3% solution, the sample is completely white without the formation of an additional phase, while in the 40% solution the precipitate is flake-like crystallites. The flakes formed with Al_p_ were completely transparent, and when we tried to examine the structure using SEM (with a probe energy of 20 kEv) at magnification, the sample began to crack. The microstructural surface was smooth (the inhomogeneities presented on it could appeared due to dislocation defects) and the 1D structures were inside the clefts. In the samples with a solution concentration of 32.6%, 1D structures were nanoscale and did not have hexagons at their base, but rather resembled needles, see Figure 7c.

Figure 8 shows the EDS data, from which the compositions on the surface of sample 5.2 and in the crack are slightly different. The macrostructure of the resulting compound based on food-grade aluminum is in a polycrystalline state. One of the reasons for the presence of crystallites is the level of purity of the laboratory room and local temperature gradients. As it was noted earlier, the reaction was violent, which is why the phase transitions may have been exothermic in nature. Dislocation transitions are clearly visible. All the resulting samples obtained from both Al_f_ and Al_p_ are stoichiometric compounds of AlF_3_. In the case of Al_f_, the total impurity is less than 1%.

After synthesis, samples 5.1 and 6.1 with the formed NWs were heated to 65 °C. The NW surface remained unchanged, indicating the thermal stability of the obtained structures.

## 4. Discussion and Conclusions

We have demonstrated the possibility of obtaining one-dimensional AlF_3_ structures under normal conditions by direct exposure of aluminum to the hydrofluoric acid in accordance with the general equation.
2Al + 6HF + 2H_2_O → 2AlF_3_·H_2_O + 3H_2_↑

Two types of aluminum alloy were investigated: food-grade aluminum and pure aluminum N_4_. The food-grade alloy exposed to HF showed an intensive reaction catalyzed by additives, while the pure aluminum reacted calmly. The analysis of the reaction products was carried out using AFM, SEM, and EDS, confirming that in most cases AlF_3_ NWs are synthesized.

The question remains on the obtained allotrope of AlF_3_. Under normal conditions, two thermodynamically stable phases of AlF_3_ are known, denoted as α and β, and some metastable phases (tetragonal and cubic). The α-AlF_3_ phase has a rhombic structure similar to that of perovskite, while the β-AlF_3_ phase has a hexagonal structure [33,34]. In general, β-MeF_3_ phases of Al, Fe and Cr are isostructural and exhibit a hexagonal tungsten bronze structure (HTB) in which MF_6_-octahedra form hexagonal channels through the structure [34]. Hexagonal NW geometry revealed by SEM points out that we are likely to have obtained β-AlF_3_ phase in our experiments. To date, there have been two main methods of β-AlF_3_ synthesis: by dehydration of α-AlF_3_∙3H_2_O and by growing crystals from chloride flux [34], so the process described in this paper may complement the set of technologies for obtaining β-AlF_3_.

We assume that a monolayer of filamentous AlF_3_ was formed, and the dendritic morphology of the resulting structure is due to the high thickness of the residue (approximately 700 microns). It cannot be said that the growth of one-dimensional structures occurred parallel to each other in view of the possible curvatures of the polypropylene substrate on which the synthesis took place. At some point, the structures must intersect and break in view of the influences exerted on each other and the presence of structural deformations. On the foundation of the already formed NWs, the synthesis of new ones begins, with which exactly the same process occurs. After some time, we observe an entire surface occupied with one-dimensional structures fused one into the other. Based on the data obtained, it is also possible to assume a decreasing dependence of the height of a NW on its diameter: the smaller the diameter is, the higher the nanowire is [35].

In order to study the mechanism, growth kinetics, and crystal structure of AlF_3_ NWs versus the conditions of our synthesis process in more detail, it is required to conduct a series of tests for growing the NWs under different supersaturations of the solution and on different substrates, for example 4H-SiC, in order to determine the evaporation rate of the solution, the corresponding rate of nucleation, and the axial growth rate of the NWs, similarly to what has been achieved for III-V semiconductor NWs [35]. To set up a stationary set of the growth parameters, it will be necessary to implement the process of compensating the solution by the film boiling rate.

When analyzing the structures using SEM, a tendency of charging of the samples was observed. From the first sample to the sixth, the charging capacity of the sample increased, due to which the obtained images had a noisy background.

Due to the easiness and simplicity of the production method, the possible production costs of AlF_3_ nanowires might be lower with the proposed technique than with the existing ones, which implies that NWs may be widely applied in sensors, high-performance electronics, and scanning probe microscopy. The obtained stoichiometric Al_p_-based AlF_3_ NWs can also be used as a dielectric in supercapacitors due to their predisposition to active charge accumulation. The robustness of the technological method under normal conditions enables the fabrication of new composite materials based on AlF_3_, obtaining multilayer structures with 1D nanocrystals with interesting electro-physical properties that can be used for the design of different microelectronic components.

Future studies will be dedicated to increasing the quality of the NW output as well as finding the optimal concentration of HF and additives conducive to the best performance of the reaction. Developing an industrial-scale technology from the shown experimental one is of certain interest as well.

## Figures and Tables

**Figure 1 nanomaterials-13-02413-f001:**
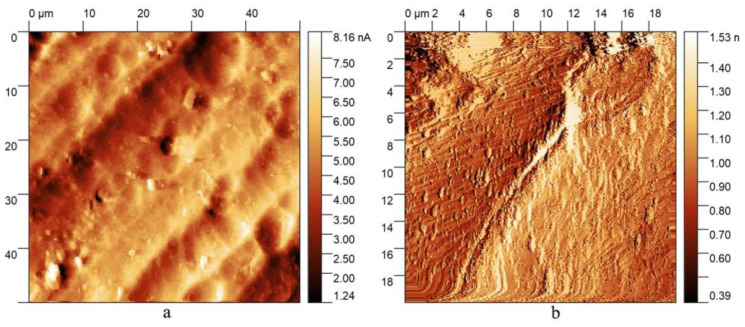
AFM images of (**a**) the initial surface of an aluminum sample; (**b**) a sample surface after the exposure to HF and the oxidation process was stopped.

**Figure 2 nanomaterials-13-02413-f002:**
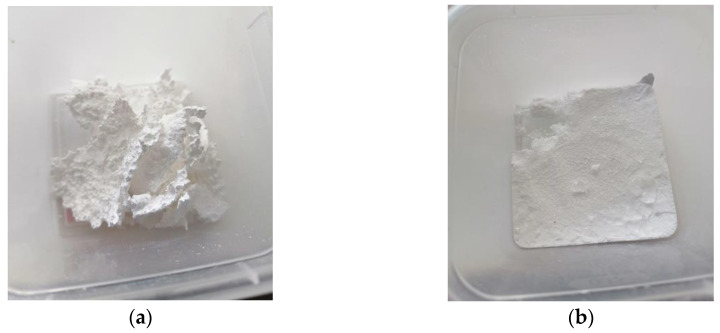
Samples obtained with different solution concentrations: (**a**) sample 2.1, concentration 32.6%; (**b**) sample 6.1, concentration 8%.

**Figure 3 nanomaterials-13-02413-f003:**
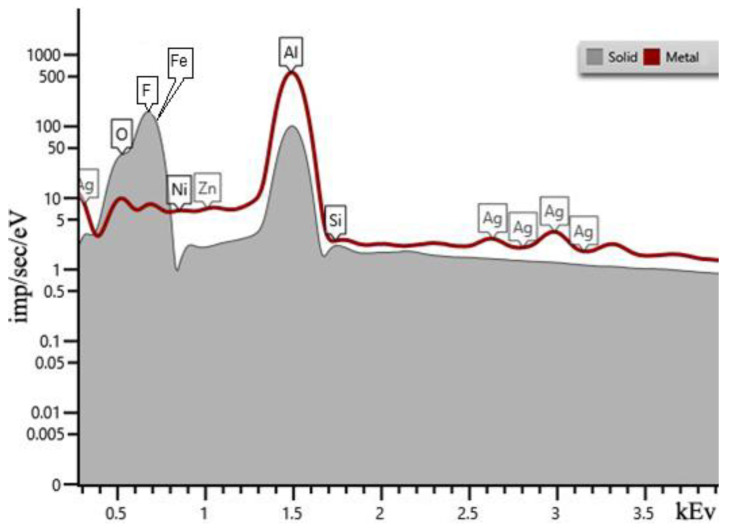
EDX spectra of sample 6.1 (“solid”). “Metal” represents Al_f_.

**Figure 4 nanomaterials-13-02413-f004:**
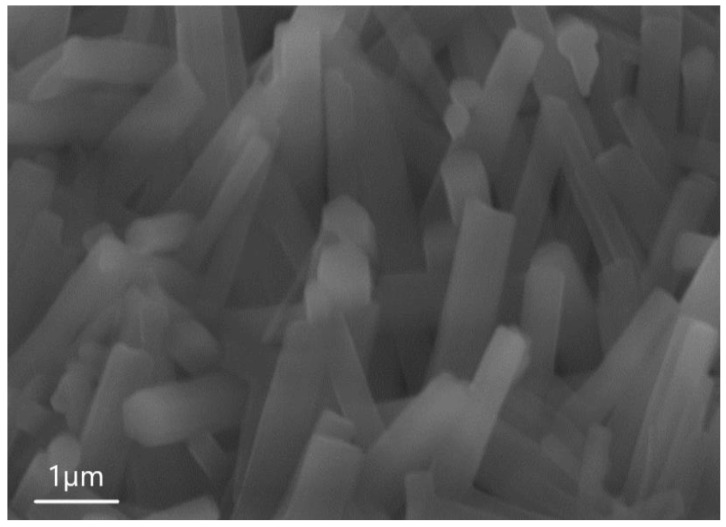
SEM image of sample 6.1, showing that the structure consists of AlF_3_ microwires.

**Figure 5 nanomaterials-13-02413-f005:**
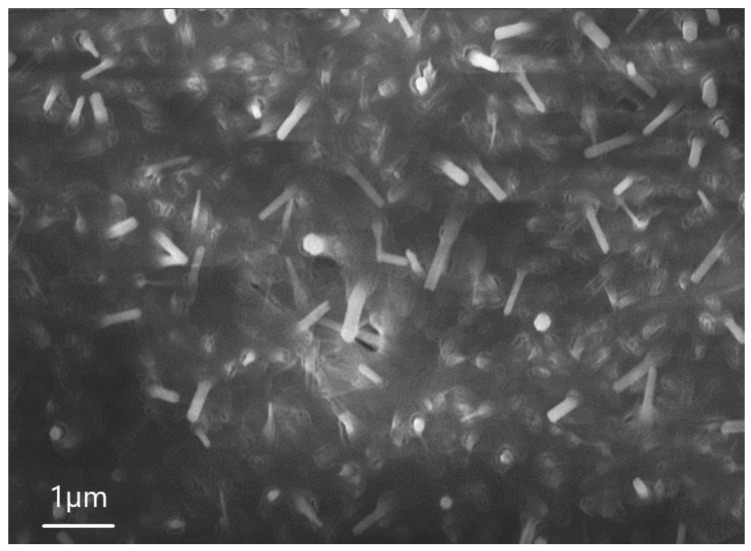
SEM image of sample 5.1.

**Figure 6 nanomaterials-13-02413-f006:**
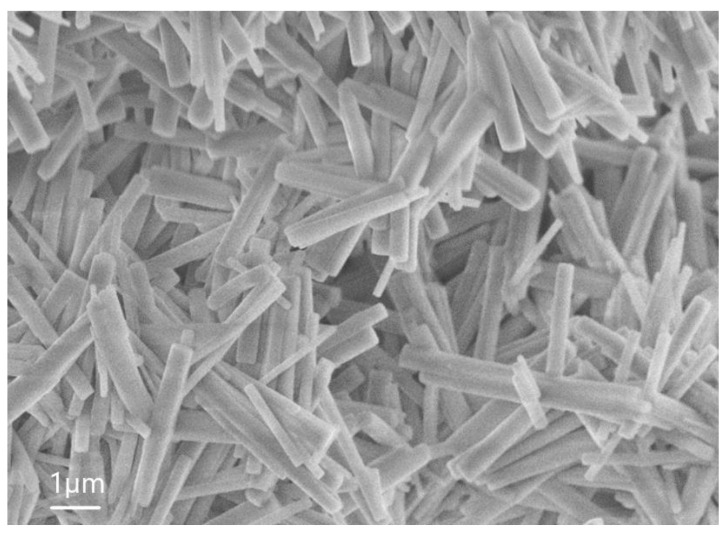
SEM image of sample 4.1.

**Figure 7 nanomaterials-13-02413-f007:**
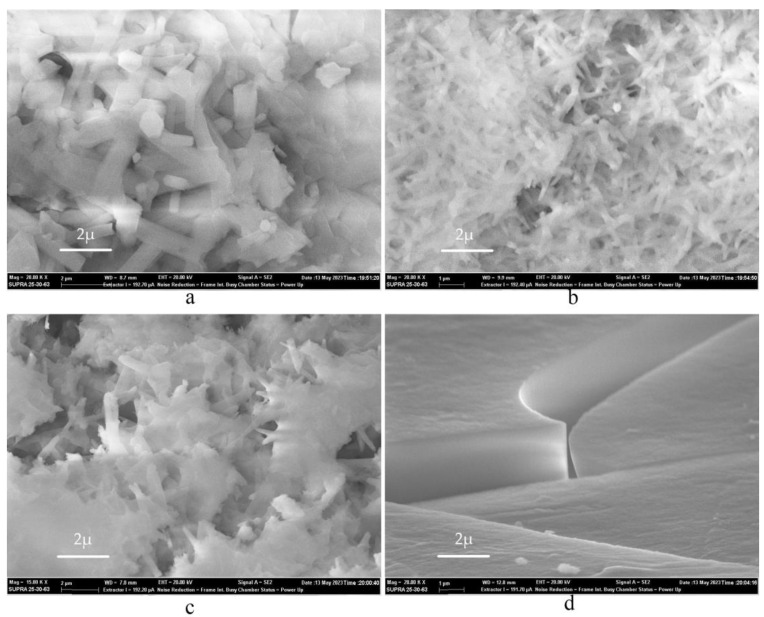
SEM image of the resulting AlF_3_ based on Al_p_, differing in HF solution concentration: (**a**) 16.3%; (**b**) 29.4%; (**c**) 32.6%; and (**d**) 40%.

**Figure 8 nanomaterials-13-02413-f008:**
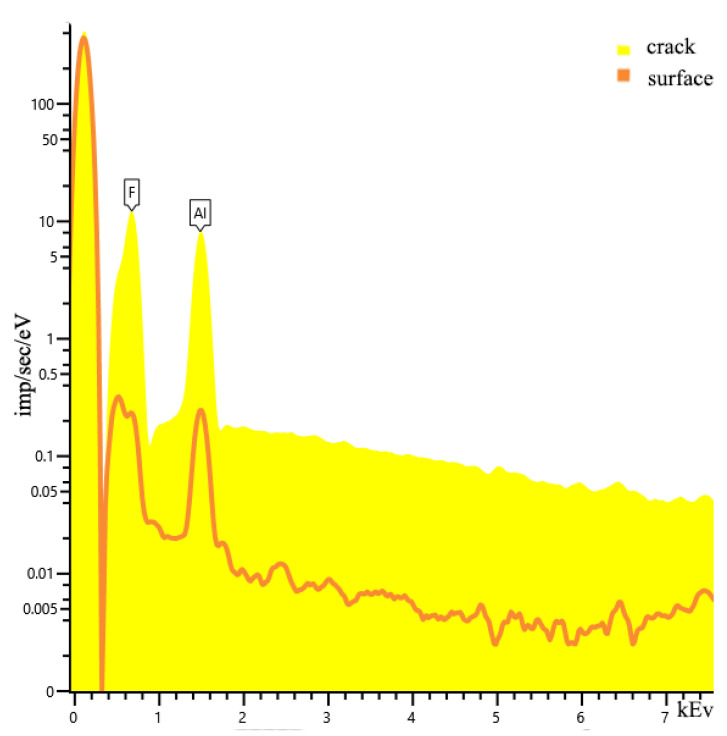
EDS data of sample 5.2.

**Table 1 nanomaterials-13-02413-t001:** Data from the tested samples of Al_f_. The residue is a resulting matter, containing AlF_3_.

Sample	Percentage of HF Solution Concentration, %	$\frac{m\left(residue\right)}{m\left(Al_{f}\right)},\;g$
1.1	40	12.49
2.1	32.6	10.67
3.1	29.4	9.77
4.1	24.5	11.34
5.1	16.3	10.12
6.1	8	14.32

**Table 2 nanomaterials-13-02413-t002:** Data from the tested samples of Al_p_.

Sample	Percentage of Solution Concentration, %	$\frac{m\left(residue\right)}{m\left(Al_{p}\right)}$
1.2	40	120.65
2.2	32.6	93.31
3.2	29.4	81.04
4.2	24.5	53.81
5.2	16.3	115.37

## Data Availability

Not applicable.

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
