# Peer review of "Synthesis under Normal Conditions and Morphology and Composition of AlF3 Nanowires"

_nanomaterials, 2023, doi:10.3390/nano13172413_

Round 1
Reviewer 1 Report
In this manuscript, the authors report on the synthesis under normal conditions, as well as on the morphology and composition of AlF3 nanowires. The subject is interesting, and the present work is timely and well presented. Synthesis technique is both original and facile, easy to reproduce. The characterization approaches (SEM, AFM, EDS) provide consistent results which are carefully analyzed and presented in all the necessary details. Consequently, the results are both original and credible. The results look credible besides being very well presented.
The figures, their captions and their corresponding discussion in the main text are easy to understand and they are logically organized too.
There are only minor concerns with this already excellent work that should be addressed before the manuscript becoming suitable for publication, i.e., it can be considered for publication after a minor revision:
1: It should be mentioned (possibly in the abstract) that a facile and controllable synthesis method has been used.
2: Abstract should briefly mention that atomic force microscopy (AFM) was adopted in the present study so it will be clear for the readers from the very beginning of the manuscript that adequate and sophisticated characterization was worked out.
3: Thermal stability of the AlF3 is an open issue. Is there any published (or preliminary) research results reporting on the (upper thermal stability limit) of AlF3 nanowires?
4: In the introduction, the authors miss that bonding and synthesis of nanowires/nanorods, etc., have already been studied (also with the purpose to understand and guide their synthesis) by using DFT and other ab initio methods, namely in [Physical Review B 68 (2003) 241401; The Journal of Physical Chemistry C 118 (2014) 5501-5509].
Spell-check and stylistic revision of the paper are necessary. Some long sentences, as well as misspellings, etc., are noticeable throughout the text.
Author Response
We would like to thank the reviewers who took their time to analyze our work and their valuable comments and criticism which we have addressed very carefully in the revision. In particular, the volume of the text has been increased, and the introduction has been expanded for clarity of the presentation and to better highlight the aspects of this work.
Below please find a point-by-point response letter, including a summary of changes made to the manuscript.
Point 1: 1: It should be mentioned (possibly in the abstract) that a facile and controllable synthesis method has been used.
Response: Thank you. We have added information that the peculiarity of our work is the simplicity and accessibility of the source materials.
Text added:
The peculiarity of our work is that the presented method for synthesis of AlF3 and one-dimensional structures based on AlF3 is simple to perform and does not require any additional precursors or costs related to the additional source materials
Point 2: Abstract should briefly mention that atomic force microscopy (AFM) was adopted in the present study so it will be clear for the readers from the very beginning of the manuscript that adequate and sophisticated characterization was worked out.
Response: A note was added in the abstract that we performed the analysis using AFM.
Text added:
We performed an intermediate atomic force microscope analysis of dissolved Al samples to analyze the reactions occurring on the metal surface.
Point 3: Thermal stability of the AlF3 is an open issue. Is there any published (or preliminary) research results reporting on the (upper thermal stability limit) of AlF3 nanowires?
Response: We did not find any mention of thermal stability of one-dimensional structures based on AlF3 in the literature, and this is an interesting topic for the future research. However, during the work we believed that SiF2 could appear during the formation of AlF3 in the reaction with Alf, and therefore an additional heat treatment was carried out at 65 oC, after which the structure remained unharmed.
Text added:
After synthesis, samples 5.1 and 6.1 with the formed NWs were heated to 65 oC. The NW surface remained unchanged, indicating thermal stability of the obtained structures.
Point 4: In the introduction, the authors miss that bonding and synthesis of nanowires/nanorods, etc., have already been studied (also with the purpose to understand and guide their synthesis) by using DFT and other ab initio methods, namely in [Physical Review B 68 (2003) 241401; The Journal of Physical Chemistry C 118 (2014) 5501-5509].
Response: Thank you. We added the following discussion in the concluding section:
Text added:
In order to study the mechanism, growth kinetics, and crystal structure of AlF3 NWs versus the conditions of our synthesis process in more detail, , it is required to conduct a series of tests for growing the NWs under different supersaturations of the solution and on different substrates, for example 4H-SiC, in order to determine the evaporation rate of the solution, the corresponding rate of nucleation and axial growth rate of the NWs, similarly to what has been achieved for III-V semiconductor NWs [36]. To set up a stationary set of the growth parameters, it will be necessary to implement the process of compensating the solution by the film boiling rate.
[36] Dubrovskii, V. G. Theory of VLS Growth of Compound Semiconductors, In: A. Fontcuberta i Morral, S. A. Dayeh and C. Jagadish, editors, Semiconductors and Semimetals, v. 93, Burlington: Academic Press, 2015, pp. 1-78.
Reviewer 2 Report
The Authors of the manuscript present a simple method for obtaining of AlF3 nanostructures using one-step chemical reaction under normal conditions. While the development of simple methods for obtaining the AlF3 nanowires is crucial for their further practical applications, and the present in this work method will be interesting for the research community dealing with the synthesis and applications of AlF3 material, the characterization of the obtained in this work nanostructures and analysis of the growth mechanisms are insufficient and must be improved before the publication of the manuscript.
I recommend the publication of this manuscript after major revision.
The major concerns are the following:
1. In the Introduction, more detailed/correct information should be provided on the properties and applications on AlF3 material to make it clear for the readers. It is only briefly mentioned in the Introduction that this material is favourable for many applications due to its “electrophysical properties”, however, no more detailed information is provided on what these properties are. In addition, for example, in the lines 36-37 is stated that “AlF3 can be used as a film to cover lithium-sulfur batteries to prevent the “shuttle effect”….”. This statement lacks the explanation, what exactly should be covered with the AlF3: the battery case, the electrode surface, what exactly? Moreover, this information is not directly related to the topic of the manuscript as the manuscript aims to develop AlF3 nanowires, not thin films.
2. In the Materials and Methods section, the description of the synthesis process is not scientific and lacks important details. For example, in line 64 is stated that “A sample of mass m = 2.435 g was placed….”, in line 66 – “the entire area of the sample was covered…..”, but nowhere is explained, what is this “sample”? What material, in what form – powder, pellets, foil?
3. The Materials and Methods section partly contains the information which is more appropriate for the Results and discussion section.
4. The presentation of the results in the Results and Discussion section is rather chaotic. The results presentation lacks systematic analysis of the obtained nanostructures in relation to the concentration of solution in terms of growth mechanisms. In addition, lines 97-99 refer to the AFM investigation, however, no AFM images or height profiles are provided anywhere in the manuscript. Some statements in the manuscript are contradictive and need more clarification, for example, in the lines 133-134 is stated that “The analysis revealed that the percentage of Zn in the microwires was higher than 133 in the matrix as a whole.”. However, the information in the Materials and methods claims that the presence of Zn in the raw material was 0.02%, but in the resulting nanostructures – 0.01% (line 121). Also, the manuscript lacks information on why the presence of Zn is important for the growth of the nanowires and what role does it play.
5. In lines 142-145, the Authors discuss the alpha and beta phases of the structure of AlF3. However, to reveal the crystalline structure, the XRD analysis of the obtained samples should be performed as SEM images do not provide sufficient information on the crystalline structure of the samples.
6. In the text, the Authors highlight the tendency of charging of the samples during the SEM investigation. Normally, this effect is observed for the nonconductive or contaminated with organic materials samples and no scientific conclusions can be drawn out of this observation except that the samples are nonconductive or contaminated.
7. The interpretation of some SEM images is not convincing. For example, regarding the SEM image in Figure 5, the Authors state that the nanowires have “a nearly perfect cylindrical shape”. However, this image shows only projection of the nanowires from the side, not their cross-section. For better understanding of the structure of the nanowires, high-resolution TEM investigation is required, which will help also to determine the crystallographic growth directions of the nanowires, as well as to reveal their crystalline structure.
8. The manuscript lacks characterization of the obtained nanowires for any of applications mentioned in the Introduction.
8. In the Conclusion section, the discussion on application of AlF3 material is included, which is not supported by the results of this work and is more appropriate for the Introduction section of the manuscript.
Minor revisions
Author Response
Thank you very much for reading and reviewing our manuscript. We did our best to improve the paper. Please, find our point-by-point reply as follows:
Point 1: In the Introduction, more detailed/correct information should be provided on the properties and applications on AlF3 material to make it clear for the readers. It is only briefly mentioned in the Introduction that this material is favourable for many applications due to its “electrophysical properties”, however, no more detailed information is provided on what these properties are. In addition, for example, in the lines 36-37 is stated that “AlF3 can be used as a film to cover lithium-sulfur batteries to prevent the “shuttle effect”….”. This statement lacks the explanation, what exactly should be covered with the AlF3: the battery case, the electrode surface, what exactly? Moreover, this information is not directly related to the topic of the manuscript as the manuscript aims to develop AlF3 nanowires, not thin films.
Response: Thank you. We have expanded the information provided on the properties and practical application of AlF3 in the management. We also clarified the meaning of the "shuttle effect" and the role of our material in solving this problem. We agree that this example does not touch on the topic of one-dimensional structures, however, the information provided shows that the synthesized AlF3 itself has properties that allow us to consider this material as selective in many directions.
Text added:
Metal fluorides are promising for catalysis applications [1-6], which are often used in C-H and C-F bond activation reactions or hydroarylation reactions, as ionic conductors [7] and luminescent materials [8-11], applying sol-gel technology with trifluoroacetic acid as a source of fluorine. In addition, Ref. [7] reports the atomic layer deposition of LiAlF4 film, which has high conductivity, homogeneity, chemical stability and electrochemical inertness. AlF3-based glasses demonstrate the enhanced chemical and thermal stability, excellent moisture resistance and are successfully used in infrared lasers [12].
Due to its electro-physical properties, it finds applications as a catalyst in organic synthesis, as an additive for electrolyte for lowering melting point, as flux for oxide separation and removal [13], as an element of multilayer optical structures [14]. It also exhibits excellent properties as a material for fabrication of far-infrared mirrors [15] in which this trifluoride is used as a protective layer for Al mirrors. In multilayer optical structures, AlF3 is a component of the AlF3/ThF4 heterostructures, which are superior to the LiF/ ThF4 optical structures in terms of their optical parameters.
AlF3 can be used as a film to cover the electrode surfaces of lithium-sulfur batteries to prevent the "shuttle effect" resulting in the transfer of polysulfide and formation of dendrites. This leads to a sharp decrease in productivity and low Kuln efficiency [19]. The use of AlF3 as a surface coating improves the electrochemical characteristics of K1.39Mn3O6 microspheres, which, in turn, has a beneficial effect on the cyclic stability, reversible capacity and high-speed capacity of batteries. The stabilized structure of AlF3 can play the role of a physical isolator between the active substances and electrolytes to prevent the occurrence of adverse reactions and inhibition of transition metal ions in batteries [19]. In most of these applications, an increased surface-to-volume ratio is expected to increase the performance of AlF3 material, which is why a cost-effective synthesis technology of one-dimensional AlF3 structures is highly desirable.
Point 2: In the Materials and Methods section, the description of the synthesis process is not scientific and lacks important details. For example, in line 64 is stated that “A sample of mass m = 2.435 g was placed….”, in line 66 – “the entire area of the sample was covered…..”, but nowhere is explained, what is this “sample”? What material, in what form – powder, pellets, foil?
Response: Thanks for the note. We clarified that the aluminum sample had a plate shape.
Text added: Two types of aluminum alloys were investigated. The first was the food-grade aluminum taken from an 1.7-liter pot by KALITVA COOK and served as a raw material for the study (Alf). In the analysis of the alloy by EDX method, the following composition was determined: Al=98.95%, Si=0.67%, Ca=0.05%, Fe=0.21%, Ni=0.13%, and Zn - 0.02%. Pieces of this raw aluminum were cut into 20x20 mm samples (plates) of 1 mm thickness, and exposed to 70 ml of the hydrofluoric acid ("Sigma Tech", Russia) of different concentrations, see Table 1.
The second aluminum alloy under investigation was the pure aluminum (Alp) in a form of pellets, with a nominal pellet mass of m = 2.435g. This aluminum of grade N4 was produced under Technical conditions ETO.021.051 and contains mass ratio of additives of 0.015% or less. In the second experiment, a single pellet was exposed to 40 ml of the hydrofluoric acid of different concentrations, see Table 2.
Point 3: The Materials and Methods section partly contains the information which is more appropriate for the Results and discussion section.
Response: The sections were restructured in response to this point. The paper now includes Introduction, Experimental, Results, and Discussion and conclusions sections.
Point 4: The presentation of the results in the Results and Discussion section is rather chaotic. The results presentation lacks systematic analysis of the obtained nanostructures in relation to the concentration of solution in terms of growth mechanisms. In addition, lines 97-99 refer to the AFM investigation, however, no AFM images or height profiles are provided anywhere in the manuscript. Some statements in the manuscript are contradictive and need more clarification, for example, in the lines 133-134 is stated that “The analysis revealed that the percentage of Zn in the microwires was higher than 133 in the matrix as a whole.”. However, the information in the Materials and methods claims that the presence of Zn in the raw material was 0.02%, but in the resulting nanostructures – 0.01% (line 121). Also, the manuscript lacks information on why the presence of Zn is important for the growth of the nanowires and what role does it play.
Response: Thank you very much for your note. We agree the analysis as a whole needs to be improved, as well as the method of conducting experiments. This work shows a possibility to synthesize one-dimensional structures of AlF3 under normal conditions, however, the ongoing experiments need to be improved in order to achieve the repeatability of the results. 2) We conducted an AFM study to observe the dissolution steps at the dislocations and on the surface of the plate. These data are now shown in Figure 1. 3) The percentage of Zn of 0.01% is given for the structure area presented in Figure 3. However, upon additional analysis, it was found that the Zn content in one NW exceeds the value that was present in the original material. We added in the text that " We hypothesize that in the first case some of additional elements, particularly zinc, may have served as a catalyst to increase the rate of the aluminum dissolution reaction".

Figure 1.
Point 5: In lines 142-145, the Authors discuss the alpha and beta phases of the structure of AlF3. However, to reveal the crystalline structure, the XRD analysis of the obtained samples should be performed as SEM images do not provide sufficient information on the crystalline structure of the samples.
Response: We agree with this. In the future work, we plan to perform a more detailed analysis using XRD and other techniques. However, it is beyond the scope of this work.
Point 6: In the text, the Authors highlight the tendency of charging of the samples during the SEM investigation. Normally, this effect is observed for the nonconductive or contaminated with organic materials samples and no scientific conclusions can be drawn out of this observation except that the samples are nonconductive or contaminated.
Response: Thank you. Analysis of any possible contaminations requires additional experiments. However, the conditions under which the Alp samples were synthesized are characterized by a high level of purity: the water for diluting the solution was deionized; storage containers pretreated with 2-methanol propane followed by deionized water; aluminum is 99.99% pure; the room complies with the cleanliness class ISO:7.
Point 7: The interpretation of some SEM images is not convincing. For example, regarding the SEM image in Figure 5, the Authors state that the nanowires have “a nearly perfect cylindrical shape”. However, this image shows only projection of the nanowires from the side, not their cross-section. For better understanding of the structure of the nanowires, high-resolution TEM investigation is required, which will help also to determine the crystallographic growth directions of the nanowires, as well as to reveal their crystalline structure.
Response: We agree that TEM analysis will give more information on the crystal structure of the NWs, but TEM study is beyond the scope of this work.
Point 8: The manuscript lacks characterization of the obtained nanowires for any of applications mentioned in the Introduction.
Response: Thank you. This work is devoted to the synthesis of one-dimensional AlF3 structures. In the future, we plan to improve the synthesis methods and conduct experiments in one of the applied areas.
Point 9: In the Conclusion section, the discussion on application of AlF3 material is included, which is not supported by the results of this work and is more appropriate for the Introduction section of the manuscript.
Response: The sections were restructured as mentioned in the answer to point 3, and the discussion of applications in the concluding section was shortened.
Round 2
Reviewer 2 Report
The authors responded to almost all comments.The paper can be published without additional changes.